# Discharge care quality in hospitalised elderly patients: Extended validation of the Discharge Care Experiences Survey

Ranveig Marie Boge[1,2]*, Arvid Steinar Haugen[3], Roy Miodini Nilsen[4,5], Frøydis Bruvik[6,7], Stig Harthug[1,5]

**1** Department of Clinical Sciences, University of Bergen, Bergen, Norway, **2** Department of Medicine, Haukeland University Hospital, Bergen, Norway, **3** Department of Anaesthesia and Intensive Care, Haukeland University Hospital, Bergen, Norway, **4** Faculty of Health and Social Sciences, Western Norway University of Applied Sciences, Bergen, Norway, **5** Department of Research and Development, Haukeland University Hospital, Bergen, Norway, **6** Haraldsplass Deaconess Hospital, Bergen, Norway, **7** Department of Global Public Health and Primary Care, Bergen, Norway

* Ranveig.Boge@helse-bergen.no

**Data Availability Statement:** All relevant data are within the manuscript and its Supporting Information files.

## Abstract

### Background

The Discharge Care Experiences Survey (DICARES) was previously developed to measure quality of discharge care in elderly patients ($\geq$ 65 years). The objective of this study was to test the factorial validity of responses of the DICARES, and to investigate its association with existing quality indicators.

### Methods

We conducted a cross-sectional study at two hospitals in Bergen, Western Norway. A survey, including DICARES, was sent by postal mail to 1,418 patients 30 days after discharge from hospital. To test the previously identified three-factor structure of the DICARES we applied a first order confirmatory factor analysis with corresponding fit indices and reliability measures. Spearman's correlation coefficients, and linear regression, was used to investigate the association of DICARES scores with the quality indicators Nordic Patient Experiences Questionnaire and emergency readmission within 30 days.

### Results

A total of 493 (35%) patients completed the survey. The mean age of the respondents was 79 years (SD = 8) and 52% were women. The confirmatory factor analysis showed acceptable fit. Cronbach's α between items within factors was 0.82 (*Coping after discharge*), 0.71 (*Adherence to treatment*), and 0.66 (*Participation in discharge planning*). DICARES was moderately correlated with the Nordic Patient Experiences Questionnaire (rho = 0.49, P < 0.001). DICARES overall score was higher in patients with no readmissions compared to those who were emergency readmitted within 30 days (P < 0.001), indicating that more positive experiences were associated with fewer readmissions.

**Funding:** This work was carried out with research grants from the Western Norway Regional Health Authority (HV911936). ASH was funded by a postdoctoral research grant (HV1172) (https://helse-vest.no/Sider/default.aspx). The funders had no role in study design, data collection and analysis, decision to publish, or preparation of the manuscript.

**Competing interests:** The authors have declared that no competing interests exist.

**Abbreviations:** DICARES, The discharge care experiences survey; CAD, Coping after discharge; ATT, Adherence to treatment; PiDP, Participation in discharge planning; NORPEQ, The Nordic patient experiences questionnaire; CCI, Charlson comorbidity index; SD, Standard deviation; CI, Confidence interval.

## Conclusions

DICARES appears to be a feasible instrument for measuring quality of discharge care in elderly patients ($\geq$ 65 years). This brief questionnaire seems to be sensitive with regard to readmission, and independent of comorbidity. Further studies of patients' experiences are warranted to identify elements that impact on discharge care in other patient groups.

## Background

Hospital discharge is a complex process starting before admission where possible, or immediately after admission [1]. In recent years, modern medical treatment and cost-effective use have ensued shorter length of hospital stay and pressure on discharge of patients [2]. A variety of adverse events are related to discharge such as drug errors, hospital-acquired infections, and procedure-related complications [3], were elderly patients are particular at risk of poorer outcomes and admissions to hospital as an emergency within 30 days of discharge (emergency readmission) [4]. A vast body of literature focuses on the patient's condition, especially cognitive impairment and vulnerability, can complicate care in the discharge process [5, 6], and cause difficulties in managing post hospitalization care [2]. Vulnerability may be related to a number of challenges, such as side effects of new drugs [7], reduced mobility and increased risk of falls [8, 9], depression [10], and lack of support system [11]. Additionally, insufficient discharge documentation and poor communication may limit the patient's ability to cope with health issues after hospitalization, contributing to increased risk of adverse events [11, 12], and rehospitalisation [11].

Over the past years, special emphasis has been placed on patient participation by involving the patients and their care givers in their own care, in accordance to their individual needs, circumstances and priorities [13]. Patient participation may be described as the state when patients' themselves become the distinct starting point for all care actions [14]. The extent of patient participation is an important indicator of the quality of healthcare [14], and has been associated with improved treatment outcomes [15, 16]. However, patients and their caregivers often feel frustrated by poor preparation for their discharge to home [16, 17], or experience that they did not have an opportunity to be involved in issues they found important to influence; like medical treatment, practical conditions and the time of discharge [18–20].

Obviously, there is a need to monitor the quality of discharge care. Monitoring and measuring quality of hospital services has a long tradition. In the days of Florence Nightingale the ultimate goal of a hospital was to manage quality by monitoring and measuring care services [21]. Today, emergency readmissions is commonly used as a general quality indicator in hospitals despite its' many inherent limitations, for instance with higher readmission rates when comorbidity increases [2, 8, 22, 23]. Better tools to investigate central factors supporting quality of transitional care, including discharge from hospital to home has been called for [16]. I has been proposed that such tools, at least partly, should be based on measuring patients' experiences [24]. Combining data on patient experiences; "the sum of all interactions influenced by all interactions shaped by an organization's culture across the continuum of care" [25], and health outcomes, are essential components used to understand and to improve the quality of hospital care [26, 27]. Positive associations between patient experiences and health outcomes have been demonstrated in several studies [28], indicating that patient experience surveys may pose as an appropriate quality indicator. Instruments measuring health condition [29, 30], comorbidity [31], and healthcare quality have been developed and validated for in-hospital use

and use after hospitalisation [26]. However, discharge care covers a variety of tasks that may influence the patients' self-care capability after hospitalisation [18, 32–35]. Hence, tools for measuring discharge care quality should have the potential to mirror how these tasks are performed by health care professionals by including questions related to important issues patients may experience after hospitalisation. Such instruments need to primarily reflect the quality of the care process rather than health conditions and comorbidity.

In a previous study we developed a patient experience instrument to measure the quality of discharge care in elderly patients (≥65 years) named as the Discharge Care Experiences Survey (DICARES) [36]. The first version comprised 10 items reflecting three factors related to discharge care: *Coping after discharge*, *Adherence to treatment*, and *Participation in discharge planning* [2, 35, 37]. The aim of this study was to investigate the DICARES' psychometric properties, and its previously identified factor structure, in a slightly modified survey. The psychometric properties and the factor structure were confirmed.

## Methods

A cross-sectional survey was conducted at two hospitals in Bergen, Western Norway, situated within the same regional health authority trust. The largest hospital is a referral tertiary teaching hospital with all specialities and covers about one million inhabitants. The smaller non-commercial private community hospital covers emergency functions for a population of approximately 150,000 inhabitants. The patients were recruited from a 22-bed internal medicine ward specialised in gastroenterology at the largest hospital, and a 32-bed general internal medicine ward at the community hospital. The distribution of patients with diseases of the digestive system at the specialized gastroenterology ward versus the general internal medicine ward was 48% and 18%, respectively. In the planning phase of our study the protocol was discussed with the hospital patient representative committee. Patient representative also participated in the study's reference group.

### Data collection

A survey was sent by postal mail to 1,418 patients ≥65 years hospitalized more than 24 hours approximately 30 days after discharge from hospital between June 2015 and April 2016. After three weeks non-responders received a reminder by mail.

The survey comprised 11 DICARES items [36], and six validated items of the Nordic Patient Experiences Survey (NORPEQ) [38, 39]. NORPEQ is commonly used as a quality indicator in Norwegian hospitals and consists of eight items designed to measure patient experiences of hospital care across the Nordic countries. The six validated items assess staff interested in problem, professional skills of nurses/doctors, nursing care, understanding doctors, and information on tests. Additionally, the survey included three questions related to patients' characteristics. Patients completing six or more DICARES-items were included in the study, corresponding to the 50% cut-off point applied in the original version of NORPEQ [38].

Data were plotted twice by the same research assistant and quality controlled for errors by two of the researchers.

### Development and previous validation of DICARES

Literature reviews, including a systematic literature review in the electronic databases PubMed, Cinahl, Embase, SweMed and PsycINFO, were conducted [36]. Our search strategy comprised the following terms: *patient experience*, *patients satisfaction*, *patient perspective*, patient *discharge*, *patient transfer*, *continuity of patient care*, *patient hand over*, *patient hand off*, *primary health care*, *home based care*, *nursing homes*, *community health services* and *community based*

*care*. In collaboration with an expert panel 16 items were extracted. Forward-translations and back-translations were conducted in order to adjust the items to fit a Norwegian context. Face validity was assessed by a group of patients, and content validity by the expert panel. The answers for each item of DICARES had five Likert-scaled choices ranging from 1 (Not at all) to 5 (To a very large extent) [40], indicating that higher scores were associated with more positive experiences. Principal component analysis identified a three factor structure comprising 10 items [36].

The previous 10-item version of the DICARES [36] was evaluated by health care professionals. Consensus was made to adjust the instrument by adding one item: *I received information about the effects and side effects of my medication*. The additional item was included due to medical care errors are one of the most commonly reported adverse events after hospitalisation [7]. The response to negative phrased items (number 1, 2, 3, 4, 9 10 and 11) were inverted to a positive scale. Minor linguistic changes were made to the current version. Principal component analysis was applied to evaluate and approve the modification (S1 File).

## Concurrent validation

We investigated concurrent validity, a type of criterion-related validity suitable for use in measuring related concepts, to examine how well DICARES correlated to two established quality indicators; the Nordic Patient Experiences Questionnaire (NORPEQ) and emergency readmission, adjusted for comorbidity. The NORPEQ- items have a five-point descriptive scale, and the NORPEQ total score is scored on a 0–100 scale from the worst experience to the best experience [38]. Emergency readmission up to 30 days to the discharging hospital was recorded from the hospitals' patient administrative system [41]. Additional information obtained from this source was age, sex, date of admission, and length of stay. Characteristics collected from the patients included educational level, housing status, and emergency readmission.

## Charlson Comorbidity Index

Charlson Comorbidity Index (CCI) [31] was used to categorize comorbidity of the patients. Each comorbidity category has an associated weight (0, 1–2, 3–4 and >5), and the sum of all the weights results in a single comorbidity score for a patient. CCI was calculated based on diagnosis codes registered by the hospitals by the International Classification of Diseases, 10th version (ICD-10) [42], and the index data were added to the dataset.

## Statistical analysis

To obtain optimal statistical power and to retain the same number of all data in the DICARES, missing data in items for a person were imputed using the mean of responses of other items for that person (within person imputation), as recommended by Siddiqui and colleagues when missing responses are ≤ 50% [43]. The differences between the non-imputed and imputed data are shown in the results, and in the supporting information files. Dependent on the distribution of the responses and the number of missing of data on each item, the mean and standard deviation may differ slightly in both directions. To obtain a measure for internal reliability for the three developed factors *Coping after discharge* (4 items), *Adherence to treatment* (3 items), and *Participation in discharge planning* (4 items), we calculated Cronbach's α. To test the factorial validity of responses of the DICARES, we applied a first order confirmatory factor analysis with the maximum likelihood estimation method [44]. Goodness of fit was assessed by use of common model fit indices with the following acceptance levels: minimum discrepancy (CMIN/df < 3.0) [45], comparative fit index (CFI ≥ 0.95) [46], root mean square

error of approximation (RMSEA < 0.06) [46], and standardised root mean square residual (SRMR < 0.05) [44]. To examine the relation between DICARES and its factors with NOR-PEQ and other characteristics, we used Spearman's correlation coefficient (rho). For this analysis, we used the total mean responses of DICARES and NORPEQ, i.e., we summarized the individual responses over the relevant items, and then divided this sum on the number of items for that scale. This was also done for the three factors of DICARES, e.g., the responses of the four items of factor *Coping after discharge* for each individual were summarized and then averaged on 4. Correlation values between 0.30 and 0.49 were considered to be satisfactory [47]. Finally, we evaluated the association of the DICARES scale and its factors with the established hospital quality indicator emergency readmission within 30 days (yes/no). This was done using DICARES and its factors as dependent variables and readmission as a dichotomous independent variable in a simple linear regression model. The analysis was repeated also after controlling for patient characteristics. To avoid list-wise deletion of individuals with missing patients' characteristics and NORPEQ responses in the adjusted analysis, we used a multiple imputation technique. We created 200 imputed datasets and the imputation model included all variables that were included in adjusted regression model. Statistical analyses were performed by Stata SE version 15 (StataCorp, College Station, Texas), SPSS version 23.0 (IBM Corp., Armonk, NY), and AMOS version 23.0 (IBM SPSS, Chicago). All P-values were two sided and values P < 0.05 were considered statistically significant.

## Ethics

This study was conducted in accordance with the Helsinki Declaration, and was approved by the Western Norway Regional Committee for Medical and Health Research Ethics (Ref.: 2015/329). A declaration of consent was attached to the survey. Patients who returned the survey with a signed consent form were included in the study. We obtained anonymous patient characteristics for all invited patients at group level from the patient administrative system. Data from the survey were stored in a designated research server at the hospital, whereas the anonymised forms were stored in a lockable cabinet according to hospital regulations.

## Results

In all, 493 (35%) patients returned questionnaires eligible for further analysis (Fig 1). Sample characteristics are shown in Table 1. The mean age was 79 years, 52% were women, 44% had a single household, and 21% reported to have obtained higher education (high school or university). The mean length of hospital stay was 3.6 days, 25% of the participants were readmitted to the hospital within 30 days, and mean score on the CCI was 0.9 (SD = 1.4). The difference in readmission rate between the two hospital wards was insignificant (P = 0.865).

Frequency and mean item responses of the 11 DICARES items for the study sample are shown in Table 2. Missing values for single items was 4.9%. Imputing person mean for missing item response did not markedly change the means or SD for any of the items.

Cronbach's α, calculated using imputed data, was estimated to be 0.82 for *Coping after discharge* (4 items), 0.71 for *Adherence to treatment* (3 items), and 0.66 for *Participation in discharge planning* (4 items) (S2 File). Confirmatory factor analysis verified satisfactory fit of the three-factor structure of the DICARES (Fig 2): CMIN/df 2.45, CFI 0.97, RMSEA 0.055 (90% CI = 0.041, 0.068) and SRMR 0.048.

Estimation of Spearman's correlation coefficient, based on imputed data, showed a moderate relationship between the DICARES factors (S3 File): *Coping after discharge* vs *Participation in discharge planning* (rho = 0.38, P < 0.001), *Participation in discharge planning* vs *Adherence to treatment* (rho = 0.40, P < 0.001), and *Coping after discharge* vs *Adherence to treatment*

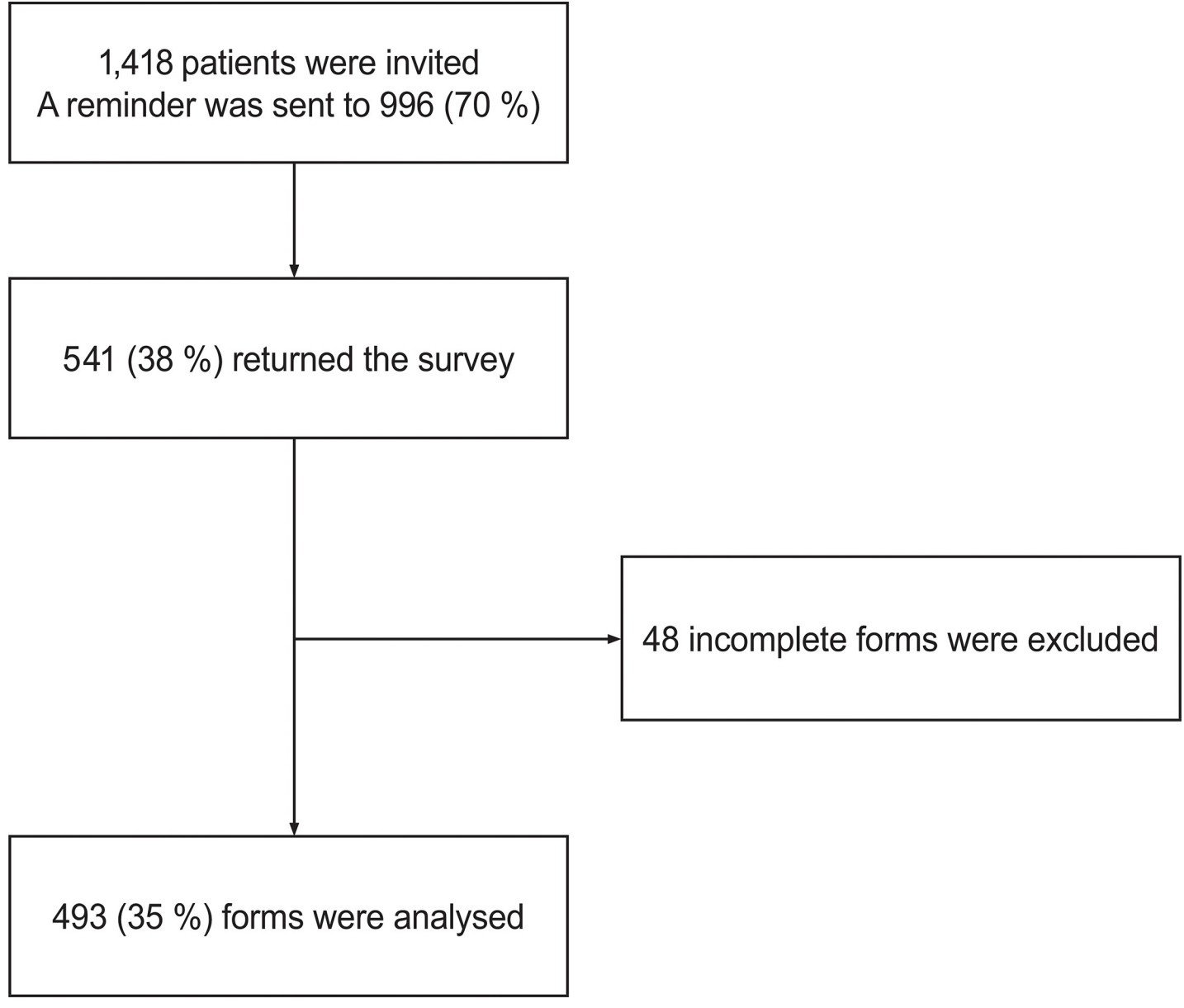

**Fig 1. Inclusion of participants in the study.** Elderly patients (≥65) were recruited from two hospitals in Bergen, Western Norway, situated within the same regional health authority. Data collection: June 2015 to April 2016.

(rho = 0.49, P < 0.001). DICARES overall (11 items) correlated moderately with NORPEQ (6 items) (rho = 0.49, P < 0.001). Correlations between the two of the three DICARES factors and NORPEQ were somewhat smaller: *Adherence to treatment* vs NORPEQ (rho = 0.40, P < 0.001), and *Coping after discharge* vs NORPEQ (rho = 0.34, P < 0.001), while there was a moderate correlation between factor *Participation in discharge planning* and NORPEQ (rho = 0.51, P < 0.001). DICARES overall, and the three factors, correlated inversely with age and had no correlation with CCI (S3 File).

The relations of scores on DICARES, and the three factors, with readmission within 30 days are shown in Table 3. Patients who were readmitted to the hospital had lower mean response than those who were not readmitted to the hospital for all factors, as well as for

**Table 1. Characteristics of the study sample.**

| Characteristics, categorical | | n | % |
|---|---|---|---|
| All patients | | 493 | 100 |
| Sex | | | |
| | Female | 257 | 52.1 |
| | Male | 236 | 47.9 |
| Patient's age, *years* | | | |
| | 65–75 | 195 | 39.6 |
| | 76–85 | 187 | 37.9 |
| | 86–99 | 111 | 22.5 |
| Household | | | |
| | Single household | 218 | 44.2 |
| | Shared household | 266 | 54.0 |
| | Missing | 9 | 1.8 |
| Education | | | |
| | Primary school | 189 | 38.3 |
| | High school low | 167 | 33.9 |
| | High school high /University | 105 | 21.3 |
| | Missing | 32 | 6.5 |
| Hospital discharge | | | |
| | Haukeland University Hospital, Bergen, Norway | 207 | 42.0 |
| | Haraldsplass Deaconess Hospital, Bergen, Norway | 286 | 58.0 |
| Emergency readmission [a] | | | |
| | No | 372 | 75.5 |
| | Yes | 121 | 24.5 |
| **Characteristics, continuous** | | **Mean** | **SD** |
| Age, *years* | | 78.5 | 8.27 |
| Charlson Comorbidity Index | | 0.93 | 1.36 |
| Length of hospital stay, *days* [b] | | 3.59 | 3.29 |
| NORPEQ [c] | | 4.03 | 0.66 |

Abbreviations: SD = standard deviation; NORPEQ = Nordic Patient Experiences Questionnaire

[a] Emergency readmitted within 30 days after discharge

[b] Data was missing for 4 patients on length of hospital stay

[c] Data was missing for 2 patients on the *Nordic Patient Experiences Questionnaire*

DICARES overall. The difference was upheld even after controlling for patient characteristics. Notably, no relation of NORPEQ with readmission was observed in unadjusted or adjusted analyses.

## Discussion

This study tested the factor structure of the DICARES, developed for monitoring discharge care quality. We found the confirmatory factor analysis to support the three factor structure; *Coping after discharge*, *Adherence to treatment* and *Participation in discharge planning*. We observed that DICARES' correlated moderately with the NORPEQ–questionnaire [38, 39]. This finding indicates that DICARES' reflects some similar aspects as the NORPEQ, and further, provide additional knowledge particularly related to discharge care quality. We found that patients with more positive experience scores on the DICARES had significantly fewer readmissions. The DICARES did not correlate with comorbidity, as measured by the CCI.

**Table 2. Item, factor, and total mean scores of the Discharge Care Experiences Survey.**

| | Respondents | Number of scores (valid %) | | | | | | With imputation of missing data [a] | |
|---|---|---|---|---|---|---|---|---|---|
| | n (%) | 1 | 2 | 3 | 4 | 5 | Mean (SD) | n | Mean (SD) |
| **Item scores** | | | | | | | | | |
| 1. I have felt stressed [b] | 488 (99) | 14 (3) | 32 (7) | 82 (17) | 138 (28) | 222 (45) | 4.07 (1.07) | 493 (100) | 4.06 (1.07) |
| 2. I have felt blue[b] | 493 (100) | 18 (4) | 47 (9) | 127 (26) | 110 (22) | 191 (39) | 3.83 (1.15) | 493 (100) | 3.83 (1.15) |
| 3. I have experienced problems in performing daily activities (e.g. personal hygiene, getting dressed or cooking) [b] | 488 (99) | 46 (9) | 35 (7) | 79 (16) | 90 (19) | 239 (49) | 3.90 (1.33) | 493 (100) | 3.90 (1.33) |
| 4. I have experienced problems in getting sufficient nutrition [b] | 488 (99) | 21 (4) | 42 (9) | 90 (18) | 62 (13) | 273 (56) | 4.07 (1.21) | 493 (100) | 4.07 (1.21) |
| 5. In connection with being discharged, I had an opportunity to notify hospital personnel about what I thought was important | 445 (90) | 57 (13) | 53 (12) | 96 (22) | 161 (36) | 78 (17) | 3.34 (1.26) | 493 (100) | 3.41 (1.23) |
| 6. When I was discharged from hospital, I understood thoroughly the purpose of taking my medication | 428 (87) | 45 (11) | 21 (5) | 43 (10) | 125 (29) | 194 (45) | 3.94 (1.30) | 493 (100) | 3.94 (1.24) |
| 7. I got information about effects and side effects of my medications | 432 (88) | 141 (33) | 84 (19) | 76 (17) | 72 (17) | 59 (14) | 2.59 (1.43) | 493 (100) | 2.79 (1.46) |
| 8. When I was discharged from hospital, I had a good understanding of my responsibility in terms of looking after my health | 478 (97) | 32 (7) | 40 (8) | 112 (23) | 203 (43) | 91 (19) | 3.59 (1.09) | 493 (100) | 3.59 (1.08) |
| 9. I have experienced problems in understanding the instructions I received when I was discharged from hospital [b] | 472 (96) | 15 (3) | 15 (3) | 32 (7) | 101 (21) | 309 (66) | 4.43 (0.98) | 493 (100) | 4.38 (1.00) |
| 10. I have experienced problems in following the instructions I received when discharged from the hospital [b] | 464 (94) | 12 (3) | 15 (3) | 37 (8) | 88 (19) | 312 (67) | 4.45 (0.95) | 493 (100) | 4.38 (0.99) |
| 11. I felt I was discharged too early [b] | 484 (98) | 27 (6) | 34 (7) | 53 (11) | 78 (16) | 292 (60) | 4.19 (1.21) | 493 (100) | 4.18 (1.21) |
| **Factor mean scores** | | | | | | | | | |
| Factor CAD (Item 1,2,3 and 4) | 493 (100) | | | | | | 3.97 (0.97) | 493 (100) | 3.97 (0.96) |
| Factor ATT (Item 5,6 and 7) | 493 (100) | | | | | | 4.34 (0.86) | 493 (100) | 4.31 (0.85) |
| Factor PiPD (Item 8,9,10 and 11) | 493 (100) | | | | | | 3.38 (0.93) | 493 (100) | 3.43 (0.89) |
| **Total mean scores** | 493 (100) | | | | | | 3.85 (0.73) | 493 (100) | 3.87 (0.71) |

Abbreviations: SD = Standard deviation; CAD = Coping after discharge; ATT = Adherence to treatment; PiDP = Participation in discharge planning

[a] Person mean imputation.

[b] Negative statements were inverted to a positive scale.

The measured indicators CMIN/df, CFI, RMSEA and SRMR showed that the hypothesized factor structure was very well adapted to the data [45, 46]. We compared the DICARES with a large inpatient care quality study by Smirnova and colleagues from 2017 [48], that in contrast to the NORPEQ-study [39], applied confirmatory factor analysis. The study included nearly 23,000 participants, were half of the respondents were > 65 years. The mean values of the sub-scale *Information at discharge* were 0.7 (scale from 0 to 1) and almost identical to the mean total DICARES score (3.85 on a scale from 1 to 5), corresponding to 70% and 71% of the respective maximum values [48]. We believe these similarities support the acceptability of DICARES in terms of being useful as an additional instrument to measure hospital discharge quality. Elderly are considerable consumers of hospital care [49] and the DICARES was developed particularly to survey experiences in this vulnerable patient group, unlike the NORPEQ [38, 39].

In a systematic review Beattie and colleagues identified 11 instruments measuring patient experience of healthcare quality [26]. We were not able to find that the instruments covered questions related to patients experience the first period after hospitalisation. Additionally, differences in methodology and timing limited comparison with the DICARES [36]. We included NORPEQ as one of the comparators in the current study since it is an established general

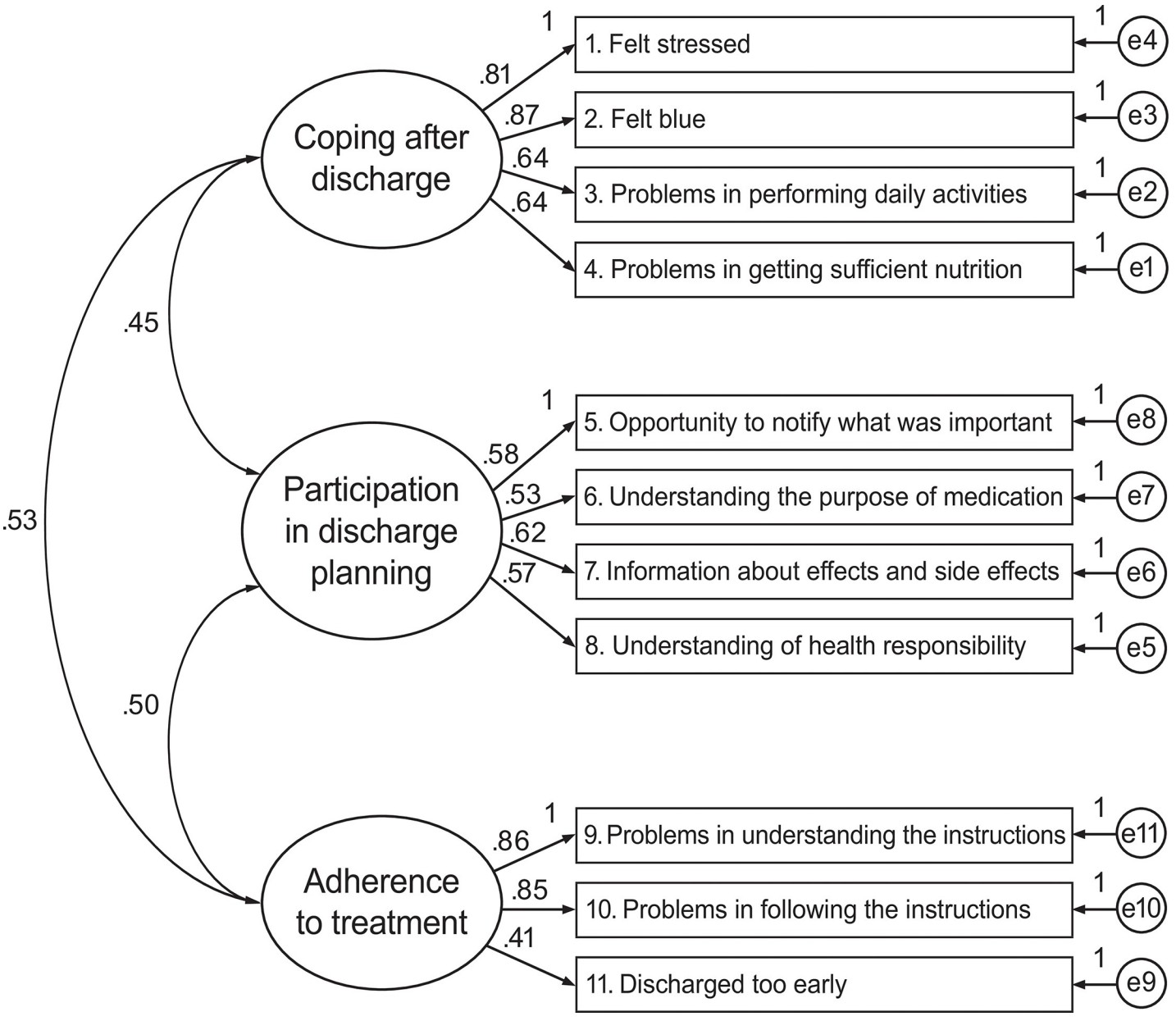

**Fig 2. Confirmatory factor analysis of the Discharge Care Experiences Survey.** Elderly patients (≥65) were recruited from two hospitals in Bergen, Western Norway, situated within the same regional health authority. Data collection: June 2015 to April 2016.

quality indicator used in Norwegian hospitals [26]. NORPEQ and Smirnova claim their instruments reflect the quality of care. This is attributed to variation in the results between or within organisations and at different organisational levels [39, 48]. Such an approach has been discussed by Bezold [50], who claims that quality will then be measured from an institutional level rather than through the eyes of the patient. Our approach has been to measure discharge care quality by comparing the DICARES with external instruments covering conditions of importance for the patients in order to identify how underlying issues may reflect specific areas of discharge.

**Table 3. Difference in total mean and factor mean scores between readmitted and not readmitted patients.**

| Scale | Emergency readmission | | | | | Estimated group difference [a] | | | | | |
|---|---|---|---|---|---|---|---|---|---|---|---|
| | No (n = 372) | | | Yes (n = 121) | | | | | | | |
| | Mean | SD | | Mean | SD | Unadjusted *b* (95% CI) | | P value | | Adjusted *b* (95% CI) [b] | P value |
| DICARES [c] | | | | | | | | | | | |
| Total (11 items) | 4.01 | 0.69 | | 3.62 | 0.74 | -0.39 | (-0.53, -0.24) | <0.001 | | -0.42 | (-0.57, -0.28) | <0.001 |
| Factor CAD (4 items) | 4.09 | 0.88 | | 3.57 | 1.10 | -0.52 | (-0.71, -0.33) | <0.001 | | -0.57 | (-0.76, -0.38) | <0.001 |
| Factor ATT (3 items) | 4.40 | 0.78 | | 4.04 | 0.99 | -0.36 | (-0.54, -0.19) | <0.001 | | -0.38 | (-0.56, -0.21) | <0.001 |
| Factor PiDP (4 items) | 3.47 | 0.91 | | 3.32 | 0.81 | -0.15 | (-0.33, 0.03) | 0.11 | | -0.20 | (-0.30, -0.01) | 0.035 |
| NORPEQ | | | | | | | | | | | |
| Total (6 items) | 4.05 | 0.67 | | 4.04 | 0.99 | -0.06 | (-0.20, 0.07) | 0.37 | | -0.09 | (-0.23, 0.04) | 0.17 |

Abbreviations: SD = standard deviation; CI = confidence interval; DICARES = Discharge Care Experiences Survey; CAD = Coping after discharge

ATT = Adherence to treatment; PiDP = Participation in discharge planning; NORPEQ, Nordic Patient Experiences Questionnaire

[a] By linear regression model

[b] Adjusted for all categorical variables in Table 1; missing data in household (n = 9), education (n = 32), and Nordic Patient Experiences Questionnaire

(n = 2) was imputed using a multiple imputation technique

[c] Missing data in items for a person were imputed using the mean of responses of other items for that person (within person imputation)

As in our previous study [36] no correlations were found between the DICARES and the CCI, indicating that comorbidity did not have a significant impact on the DICARES scores. We may have succeeded to develop an instrument that measures health service quality rather than the patients' health conditions influenced by comorbidity, in our study measured by CCI. The DICARES is simple, brief and its three factors have the potential, directly or indirectly, to reflect specific areas discharge care quality [51]. The response of each item indicates sufficient variation in the responses and normal distribution [52].

According to Manary and colleagues [53], patient experience measures do not simply reflect clinical adherence-driven outcomes, but also another dimension of quality which otherwise is difficult to measure objectively. We believe the DICARES' three-factor structure makes it possible to identify and measure underlying issues in quality of care and that suitable strategies may be developed and implemented through quality improvement work [54, 55].

In the current study we chose to use emergency readmission for concurrent validation of the DICARES. The factors *Coping after discharge* and *Adherence to treatment* were associated with readmission, indicating emergency readmission as a quality indicator, and the DICARES covers some similar aspects. This is in line with results in the study of Kangovi and colleagues who found that one of the most commonly reported issues that contributed to readmission was difficulties in performing daily tasks [34]. Factor *Adherence to treatment* was significantly lower for the readmitted patients versus the non-readmitted patients in the current study. Adherence is the primary determinant of the effectiveness of treatment and is affected by the patient-provider relationship, and also by numbers of patient-related factors such as low motivation, lack of a self-perceived need for treatment, feeling of being discharged too early from previous hospitalisation, or multiple hospital admissions [34, 54, 56–58].

Patients reported the lowest scores for the factor *Participation in discharge planning*.

This result is similar to the findings in the previous DICARES' study [20], and corresponds with elderly patients' experiences of not being involved in discharge planning from hospital [18, 59, 60]. Despite the lack of participation, elderly patients' interviewed in a study of Hvalvik and colleagues [60] were humble and expressed gratefulness for the care system they were a part of. The authors claim a patient-oriented approach as essential in the process to support the elderly patients because they are challenged during the transition between hospital and

home. To support care of elderly patients, health care professionals need to understand the patient's present situation in the context and coherence of past and future [60]. Patients with positive care experiences are often more engaged in their care, more committed to treatment plans, and more receptive to medical advices [51].

A limitation of the current study may be the relatively low response rate, though it is comparable to the study by Smirnova and colleagues [48]. Low participation is a major concern in patient experience surveys [38]. One concern could be that elderly persons with a high CCI would participate to a lesser extent. In a previous study of DICARES [36] investigating patient experiences in a similar population of elderly, the response rate was 64% and the CCI was 0.7 higher than in the current study. This indicate that comorbidity may not be the reason for the limited number of responders in the current study. However, the response rate may have been influenced by geriatric syndromes; clinical conditions that is common in elderly and that do not fall into distinct disease categories, like weight loss, pain and depressive symptoms [61]. Another limitation may be that patients who completed less than six DICARES items were not included in the study. Poor condition or cognitive impairment could be reasons for lack of completion of the questionnaire. Exclusion of these patients may have biased the results.

Unlike findings in the previous study of the DICARES [20], Cronbach's α was somewhat lower for the factor *Participation in discharge planning* than required according to quality criteria for measurements [26]. However, instruments for quality improvement may tolerate lower levels of reliability in favour of other aspects of utility, such as it is brief and there are good theoretical and practical reasons for the instrument [62] due to educational impact, cost and acceptability [26]. Measurement error is not calculated, similar to results in Beattie and colleagues systematic review where only one of the studies reported on this criterion [26]. Except from these possible weaknesses DICARES' fulfils the other quality criteria for measurement properties.

The DICARES meet with recommendations of Manary and colleagues [53] who claim that patient experiences measurement should address a specific event or visit, focus on provider patient interactions, and be assessed in a timely manner. Furthermore, the DICARES is in accordance with the usual distribution of surveys to patients in clinical improvement work. We find it important to keep the questionnaire brief, otherwise elderly sick patients may find it too demanding to complete. The survey was distributed to the patients one month after discharge as this was relevant due to comparison with the quality indicator emergency readmission within 30 days. There may be patients who did not receive the questionnaire because they were already readmitted at the time the questionnaire was sent. Further, there may be patients who did not answer the questionnaire because they had already been readmitted at that time, which may have resulted in a failure to answer the questionnaire even though a poor discharge process was the reason for re-admission. Additionally, there is a risk of recall-bias that patients who have been readmitted confuse the experiences of more admissions. However, test-retest showed satisfactory results in a previous study [36]. The CCI is limited to cover only the prognostic aspect as a risk of early mortality [31], and unlike the previous study of the DICARES [36], a health status survey is not included in this study. The amount of missing data was acceptable [63]. By applying imputation the power of the analyses has been strengthen, and the risk of bias reduced.

## Conclusions

The DICARES appears to be a valid questionnaire for measuring discharge care quality. The survey provides additional value to the knowledge of challenges faced by patients, and contributes to verify the feasibility of the DICARES. When compared with established hospital quality

indicators, the results indicate that DICARES could be a feasible tool to add to discharge improvement measures. DICARES seems to have sensitive properties with regard to the readmitted patients, and to be independent of comorbidity. The three factor structure may reflect directly and indirectly underlying issues related to discharge. The psychometric evaluation of the DICARES suggests acceptable internal consistency, and adequate construct validity of the instrument as a whole. DICARES is a brief, generic, non-diagnostic, and specific questionnaire. Further validation may also include elderly patients discharged from general surgical units.

## Supporting information

**S1 File. Principal component analysis.**
(DOCX)

**S2 File. Reliability analysis.**
(DOCX)

**S3 File. Spearman's correlation coefficient.**
(DOCX)

**S4 File. Available data.** Anonymous data set including 493 respondents.
(XLSX)

## Acknowledgments

We would like to thank the respondents for their efforts and also healthcare professionals and the Head of Departments in the participating hospitals. The authors would also like to thank Britt Elin Arnetvedt Erdal for her accurate and efficient entry of data.

## Author Contributions

**Conceptualization:** Ranveig Marie Boge, Stig Harthug.

**Formal analysis:** Ranveig Marie Boge, Arvid Steinar Haugen, Roy Miodini Nilsen, Stig Harthug.

**Investigation:** Ranveig Marie Boge.

**Methodology:** Ranveig Marie Boge, Arvid Steinar Haugen, Roy Miodini Nilsen, Stig Harthug.

**Project administration:** Ranveig Marie Boge, Stig Harthug.

**Supervision:** Arvid Steinar Haugen, Roy Miodini Nilsen, Stig Harthug.

**Writing – original draft:** Ranveig Marie Boge.

**Writing – review & editing:** Arvid Steinar Haugen, Roy Miodini Nilsen, Frøydis Bruvik, Stig Harthug.

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
