## [Decision Letter · Decision Letter 0]

15 Jul 2019

PONE-D-19-15237

Discharge care quality in hospitalised elderly patients: Extended validation of the Discharge Care Experiences Survey

PLOS ONE

Dear Ms Boge,

Thank you for submitting your manuscript to PLOS ONE. After careful consideration, we feel that it has merit but does not fully meet PLOS ONE’s publication criteria as it currently stands. Therefore, we invite you to submit a revised version of the manuscript that addresses the points raised during the review process.

We would appreciate receiving your revised manuscript by Aug 29 2019 11:59PM. To enhance the reproducibility of your results, we recommend that if applicable you deposit your laboratory protocols in protocols.io, where a protocol can be assigned its own identifier (DOI) such that it can be cited independently in the future. For instructions see: http://journals.plos.org/plosone/s/submission-guidelines#loc-laboratory-protocols

We look forward to receiving your revised manuscript.

Kind regards,

Prof, Mojtaba Vaismoradi, PhD, MScN, BScN

Academic Editor

PLOS ONE

Journal Requirements:

1. Thank you for including your ethics statement:

"Western Norway Regional Committee for Medical and Health Research Ethics (Ref.:2015/329). Written consent form were obtained."

Please amend your current ethics statement to confirm that your named institutional review board or ethics committee specifically approved this study.

2. Please upload a copy of Supporting Information Figure S4 which you refer to in your text

Reviewers' comments:

**Comments to the Author**

Reviewer #1: Summary of research

The publication "Discharge care quality in hospitalized elderly patients..." shows an interesting study on the extended validation of DICARES. The previously developed and validated questionnaire DICARES was subjected to an extended validation at 2 hospitals with different levels of care in Bergen (Norway) by means of a cross sectional design study. The patients were recruited from a specialized gastroenterological ward of a maximum care hospital and a general internal ward of a community hospital. The aim was to investigate the psychometric properties of DICARES and the previously identified structure of the factors. Testing was performed against other validated questionnaires.

The special feature of DICARES is its focus on accompanying the discharge ("discharge care"). The patients who were not re-admitted showed a higher DICARES score than the patients who were re-admitted. The study was conducted 30 days after discharge and the discharged patients received the questionnaire by post. The results show appropriately high values of Cronbach's alpha of the items within the factors "Coping after discharge", "Adherence to treatment" and "Participation in discharge planning" with decreasing values of 0.82, 0.71 and 0.66. The statistical tests carried out are comprehensibly documented and comprehensible.

The study, well conducted in an appropriate design and with appropriate methods, has demonstrated that DICARES is an applicable tool that can capture the quality of nursing preparation of the discharge process sensitively in terms of patient readmission and independently in terms of patient comorbidity.

The study thus proves the effectiveness of the instrument for the intended purpose.

Overall impression

The study meets the required ethical requirements through approval by the responsible ethics committee. The statistical analysis was carried out appropriately. An informed consent was obtained from the patients. The personal data collected was anonymized and data protection requirements for the processing and storage of the data have been fulfilled.

The study presents a well-developed and two-stage validated instrument that, with its focus on the quality of the preparation of the discharge process. The study makes an important contribution to the quality assurance of the treatment process at the interface inpatient to outpatient.

Overall comment

The question I am thinking about while I’m working out this peer-review is whether the fact that the questionnaire was sent 30 days after the discharge, but the readmission rate was evaluated within those 30 days, could have biased the results, and in what extent.

The questionnaires were sent at day 30 after discharge. At the same time, however, the patient readmissions were evaluated within 30 days. The study provides no information whether and how many patients who were readmitted within 30 days (especially early after discharge) got and/or completed the questionnaire.

It is correct to focus on an established hospital quality indicator such as "readmission within 30 days", as this is easy to measure. There may be patients who did not receive the questionnaire because they were already readmitted at the time the questionnaire was sent. And there may be patients who did not answer the questionnaire because they had already been re-admitted at that time, which may have resulted in a failure to answer the questionnaire even though a poor discharge process was the reason for re-admission. Despite these possible dropouts, however, the results obtained are to be regarded as correct and meaningful, as the suitability of the questionnaire was determined.

To address these thoughts, a table, graph or plot of the data showing which patients (proportions) got the questionnaire, and which answered the questionnaire depending of the status “readmitted” could be helpful.

It would be interesting to know a textual or other description of the extent to which the results differ in relation to the level of care (specialized ward / community ward).

Maybe – because this study provides a solid answer to the question to be dealt with – it would be interesting to conduct another study in which the questionnaire will be given within the first few days (e.g. 3 or 5 days) after discharge to see if the results of the questionnaire can predict the risk of a readmission.

Overall Recommendation

The study raises a small number of questions, some of which can be resolved by simple explanations and clarifications in the text.

Clarification should be provided on the issue of imputation and the corresponding presentation of results and textual presentation in order to remove minor ambiguities.

The planned extended validation of DICARES was carried out properly and the effectiveness and value of the instrument was proven.

The Issues to address should be worked out. The study should be published.

Issues to address

Line 98-100: The patients were recruited from a specialized gastroenterological ward of a maximum care hospital and a general internal ward of a community hospital.

> Consideration should be given in the text to whether the patient population was comparable (enough), maybe with just a few sentences. Since the wards differ in terms of specialization, the more specialized ward may have had lower readmission rates than the less specialized ward or vice versa.

Maybe consideration should be given on an adjustment of the analysis of the data depending on the level of specialization of the ward, but maybe this will be worked out in the further analysis planned as addressed in the article.

Lines 134 ff and 155-159: For questions addressing the imputations of data see Issues Line 201 (Table 2) and Supporting information (File 2) as follows.

Line 201 (Table 2) and Supporting information (File 2): Table 2 shows information and results for calculations with imputations for missing data. The mean values with the imputations deviate in both directions (+/-) from the mean values without the imputations or remain stable. The SD sometimes gets narrower, wider or remains stable, too. Thus, it becomes clear that the imputations have an influence on the mean values shown.

> The meaning of the change in the mean values due to the imputations should be explained or presented in the text.

The total mean score is only calculated for the data without imputations.

> The total mean score in Table 2 should also be calculated for the column with the imputations.

> It should be clearly stated in the text on which mean score given in table 2 (with or without imputations) the further statements are based, as the imputations could have distorted the results.

The File 2 provides a kind of sensitivity-analysis by calculating the average covariance and Cronbach's alpha for the data with and without imputations.

> A representation of the mean scores as in File 2 separated into data with and without imputation should be checked.

Minor issues

References: The DOI of Reference No. 28 is not correct, the article could not be retrieved via that DOI. All References should be checked, I didn’t check them systematically.

Line 90/91: The sentence “The psychometric…” impresses as a part of the results.

The sentence should be moved to the results section or reworded.

Line 108/109: Only patients who completed more than 50% of the DICARES-Items were included in the study.

This criterion led to the exclusion of feedback forms that might have indicated a link between discharge preparation and readmission. The reason for the decision to set this 50% cut-off should be given in the text, as should the possible effects of this decision. Any kind of re-analysis of the data regarding this aspect seems not to be necessary.

Line 124: the word “patients” in the sentence “…, length of stay and patients” seems to be out of context, as long it does not address the height of the patient

Line 176: the percentage “53%” differs from the value given in the abstract (52%)

Line 244/245: There is an incomplete sentence “The concurrent validity of the.” If a complete paragraph or section is missing, I’d like to know what it was.

Line 276: doubled words “in order”

Line 278: wording should be “no correlation was found” or “no correlations were found”

Line 283: wording should be “reflect specific areas of discharge”

Line 294: wording should be “DICARES covers some similar aspects”

Line 296: doubled word “Factor Adherence to treatment factor was…”

Line 340: wording should be “The CCI is limited to only cover…” or “The CCI is limited to cover only…”

Line 248: wording should be “…contributes to verify…”

Line 350: wording should be “DICARES seems to have…”

Reviewer #2: General comments:

- Pay attention to punctuation, for instance line 39.

Introduction:

- The section needs major revision. The rationale for the study should be strengthened. Why is this study important to do in elderly patients? Authors must add more literature about the issue.

Methods:

- In general, this section should be more organized. Create some subheadings like Concurrent validity, Charlson comorbidity index and …, then provide a complete explanation for each of them.

- In the methods section it has been stated that development and validation of the DICARES comprised literature reviews, consultations with an expert panel, patients’ evaluation and principal component analysis. This section should be enriched with additional details such as search flow and so on.

- More information is needed about the NORPEQ.

Discussion

- Line 245, it seems to be incomplete.

---

## [Author Response · Author response to Decision Letter 0]

11 Aug 2019

We are grateful for the reviewers’ and Editor’s comments and their suggestions to improve the manuscript. We hope that our responses have addressed the comments sufficiently and that our manuscript would be acceptable for publication. 

Response to reviewers’

Reviewers’ comments

Response to reviewer #1

Overall comment

The question I am thinking about while I’m working out this peer-review is whether the fact that the questionnaire was sent 30 days after the discharge, but the readmission rate was evaluated within those 30 days, could have biased the results, and in what extent.

The questionnaires were sent at day 30 after discharge. At the same time, however, the patient readmissions were evaluated within 30 days. The study provides no information whether and how many patients who were readmitted within 30 days (especially early after discharge) got and/or completed the questionnaire.

It is correct to focus on an established hospital quality indicator such as "readmission within 30 days", as this is easy to measure. There may be patients who did not receive the questionnaire because they were already readmitted at the time the questionnaire was sent. And there may be patients who did not answer the questionnaire because they had already been re-admitted at that time, which may have resulted in a failure to answer the questionnaire even though a poor discharge process was the reason for re-admission. Despite these possible dropouts, however, the results obtained are to be regarded as correct and meaningful, as the suitability of the questionnaire was determined.

To address these thoughts, a table, graph or plot of the data showing which patients (proportions) got the questionnaire, and which answered the questionnaire depending of the status “readmitted” could be helpful.

It would be interesting to know a textual or other description of the extent to which the results differ in relation to the level of care (specialized ward / community ward).

Response: Thank you. Unfortunately, we do not have permission to obtain data from patients who did not consent to participate. However, in the Discussion we have added “There may be patients who did not receive the questionnaire because they were already readmitted at the time the questionnaire was sent. Further, there may be patients who did not answer the questionnaire because they had already been readmitted at that time, which may have resulted in a failure to answer the questionnaire even though a poor discharge process was the reason for re-admission.

Issues to address

Line 98-100: The patients were recruited from a specialized gastroenterological ward of a maximum care hospital and a general internal ward of a community hospital.

> Consideration should be given in the text to whether the patient population was comparable (enough), maybe with just a few sentences. Since the wards differ in terms of specialization, the more specialized ward may have had lower readmission rates than the less specialized ward or vice versa. 

Response: Agree. We added information to the Methods section: “The distribution of patients with diseases of the digestive system at the specialized gastroenterology ward versus the general internal medicine ward was 48% and 18%, respectively. (line 124-126)

We also added information to the Results section: “The difference in readmission rate between the two hospital wards was insignificant (P=0.865).” (line 231-232)

Maybe consideration should be given on an adjustment of the analysis of the data depending on the level of specialization of the ward, but maybe this will be worked out in the further analysis planned as addressed in the article. 

Response: We agree, this could be interesting to examine in future studies of DICARES. 

Lines 134 ff and 155-159: For questions addressing the imputations of data see Issues Line 201 (Table 2) and Supporting information (File 2) as follows.

Line 201 (Table 2) and Supporting information (File 2): Table 2 shows information and results for calculations with imputations for missing data. The mean values with the imputations deviate in both directions (+/-) from the mean values without the imputations or remain stable. The SD sometimes gets narrower, wider or remains stable, too. Thus, it becomes clear that the imputations have an influence on the mean values shown. 

> The meaning of the change in the mean values due to the imputations should be explained or presented in the text. 

Response: Agree. Sentences added to clarify: “The differences between the non-imputed and imputed data are shown in the results, and in the supporting information files. Dependent on the distribution of the responses, and the number of missing of data on each item, the mean and standard deviation may differ slightly in both directions.” (line 186-189)

We added the following text in the Discussion section: “By applying imputation the power of the analyses has been strengthen, and the risk of bias reduced. (line 411-412)

The total mean score is only calculated for the data without imputations. 

> The total mean score in Table 2 should also be calculated for the column with the imputations. 

Response: Agree. Total mean score for the column with the imputations have been added to Table 2. Line (line 255).

> It should be clearly stated in the text on which mean score given in table 2 (with or without imputations) the further statements are based, as the imputations could have distorted the results. 

Response: Agree. Information added: “Cronbach’s α, calculated using imputed data, was estimated to be 0.82 for Coping after discharge (4 items), 0.71 for Adherence to treatment (3 items), and 0.66 for Participation in discharge planning (4 items) (S2 file).” (line 248)

“Estimation of Spearman’s correlation coefficient based on imputed data, showed a moderate relationship between the DICARES factors (S3 file),….” (line 271)

 The File 2 provides a kind of sensitivity-analysis by calculating the average covariance and Cronbach's alpha for the data with and without imputations. 

> A representation of the mean scores as in File 2 separated into data with and without imputation should be checked

Response: Agree. Mean scores with and without imputations for the three factors have been added to Table 2. (line 255)

Minor issues

References: The DOI of Reference No. 28 is not correct, the article could not be retrieved via that DOI. All References should be checked, I didn’t check them systematically. 

Response: All references are checked and corrections made. We opened DOI of reference No. 28 successfully. Reference 43 (line 574-576).

Line 90/91: The sentence “The psychometric…” impresses as a part of the results. The sentence should be moved to the results section or reworded. 

Response: We do not agree to this point with respect to Plos One’s Submission guidelines for “Introduction”: “Conclude with a brief statement of the overall aim of the work and a comment about whether that aim was achieved”. https://journals.plos.org/plosone/s/submission-guidelines#loc-abstract

Line 108/109: Only patients who completed more than 50% of the DICARES-Items were included in the study. 

This criterion led to the exclusion of feedback forms that might have indicated a link between discharge preparation and readmission. The reason for the decision to set this 50% cut-off should be given in the text, as should the possible effects of this decision. Any kind of re-analysis of the data regarding this aspect seems not to be necessary. 

Response: Agree. Text added in two sections: 

“Patients completing six or more DICARES-items were included in the study, corresponding to the 50% cut-off point applied in the original version of NORPEQ”. (line 140-141)

“To obtain optimal statistical power and to retain the same number of all data in the DICARES, missing data in items for a person were imputed using the mean of responses of other items for that person (within person imputation), as recommended by Siddiqui and colleagues when missing responses are ≤ 50%.” (line 183-186)

In discussion: “Patients who completed less than six DICARES items were not included in the study. Poor condition or cognitive impairment could be reasons for lack of completion of the questionnaire. Exclusion of these patients may have biased the results”. (line 381-382)

Line 124: the word “patients” in the sentence “…, length of stay and patients” seems to be out of context, as long it does not address the height of the patient

Response: Agree. Sentence reworded to: Additional information obtained from this source was age, sex, date of admission and length of stay. (line 170-171)

Line 176: the percentage “53%” differs from the value given in the abstract (52%)

Response: Agree. Changed to 52%. (line 228)

Line 244/245: There is an incomplete sentence “The concurrent validity of the.” If a complete paragraph or section is missing, I’d like to know what it was.

Response: Sentence removed.

Line 276: doubled words “in order”

Response: Removed.

Line 278: wording should be “no correlation was found” or “no correlations were found”

Response: Corrected. (line 334)

Line 283: wording should be “reflect specific areas of discharge”

Response: Corrected. (line 333)

Line 294: wording should be “DICARES covers some similar aspects”

Response: Corrected. (line 349-350)

Line 296: doubled word “Factor Adherence to treatment factor was…”

Response: Corrected. (line 352)

Line 340: wording should be “The CCI is limited to only cover…” or “The CCI is limited to cover only…”

Response: Corrected. (line 408-409)

Line 248: wording should be “…contributes to verify…”

Response: Corrected. (line 417)

Line 350: wording should be “DICARES seems to have…”

Response: Corrected.

Response to reviewer #2

Reviewer #2: General comments:

- Pay attention to punctuation, for instance line 39.

Response: Thank you. The manuscript has been checked with respect to punctuation.

Introduction:

- The section needs major revision. The rationale for the study should be strengthened. Why is this study important to do in elderly patients? Authors must add more literature about the issue. 

Response: We agree. More text and 17 references added. (line 61 to 64, line 66-71, line 72-73, line 75-84).

Methods:

- In general, this section should be more organized. Create some subheadings like Concurrent validity, Charlson comorbidity index and …, then provide a complete explanation for each of them.

Response: Response: Agree. Several changes made in Methods between line 124 and line 180, several subheadings added and explanation provided: 

Concurrent validation

“We investigated concurrent validity, a type of criterion-related validity that is suitable to measure related concepts, to examine how well DICARES correlated to two established quality indicators; the Nordic Patient Experiences Questionnaire (NORPEQ) and emergency readmission, adjusted for comorbidity”. (line 163-167)

Charlson Comorbidity Index

“Charlson Comorbidity Index (CCI) [16] was used to categorize comorbidity of the patients. Each comorbidity category has an associated weight (0, 1-2, 3-4 and >5), and the sum of all the weights results in a single comorbidity score for a patient”. (line 175-178)

- In the methods section it has been stated that development and validation of the DICARES comprised literature reviews, consultations with an expert panel, patients’ evaluation and principal component analysis. This section should be enriched with additional details such as search flow and so on.

Response: Agree. Changed made, and text added between line 146 and 161.

- More information is needed about the NORPEQ.

Response: Agree. Changed to: “The survey comprised 11 DICARES items[36], and six validated items of the Nordic Patient Experiences Survey (NORPEQ)[38, 39]. NORPEQ is commonly used as a quality indicator in Norwegian hospitals and consists of eight items designed to measure patient experiences of hospital care across the Nordic countries. The six validated items assess staff interested in problem, professional skills of nurses/doctors, nursing care, understanding doctors, and information on tests. (line 134-139)

Discussion

- Line 245, it seems to be incomplete.

Response: We agree. Sentence removed.

---

## [Decision Letter · Decision Letter 1]

27 Aug 2019

PONE-D-19-15237R1

Discharge care quality in hospitalised elderly patients: Extended validation of the Discharge Care Experiences Survey

PLOS ONE

Dear Ms Boge,

Thank you for submitting your manuscript to PLOS ONE. After careful consideration, we feel that it has merit but does not fully meet PLOS ONE’s publication criteria as it currently stands. Therefore, we invite you to submit a revised version of the manuscript that addresses the points raised during the review process.

We would appreciate receiving your revised manuscript by Oct 11 2019 11:59PM. To enhance the reproducibility of your results, we recommend that if applicable you deposit your laboratory protocols in protocols.io, where a protocol can be assigned its own identifier (DOI) such that it can be cited independently in the future. For instructions see: http://journals.plos.org/plosone/s/submission-guidelines#loc-laboratory-protocols

We look forward to receiving your revised manuscript.

Kind regards,

Prof, Mojtaba Vaismoradi, PhD, MScN, BScN

Academic Editor

PLOS ONE

Reviewers' comments:

Reviewer #1: All my comments from the first round of the review were adequatly adressed. There are only 4 minor issues left, that should be worked out prior to publication.

Line 81: missing "to", should be ...for their discharge to home

Line 123: should be ....medicine ward specialised in gastroenterology

Line 321: should be "In a systematic review Beattie and colleagues identified 11 instruments measuring patient experience of healthcare quality [26]." or "Beattie and colleagues identified 11 instruments of measuring patient

experience of healthcare quality in a systematic review [26]."

And - sorry for grouching, but - the DOI for Reference 43 leads me to the wrong publication, I cannot retrieve the right publication with the given DOI 10.12788/jhm.3037 via doi.org. In the citation 43 as given I can only find another publication, were the title and one authors name match.

The DOI 10.12788/jhm.3037 given in the citation 43 leads to the following publication:

Brotman DJ, Siddiqui Z, Siddiqui Z, Durkin N. Does Patient Experience Predict 30-Day Readmission? A Patient-Level Analysis of HCAHPS Data. Journal of Hospital Medicine. 2018;13. doi:10.12788/jhm.3037

The publication you cited as No. 43 has another DOI, when I look for the title only:

Siddiqui OI. Methods for Computing Missing Item Response in Psychometric Scale Construction. Current Research in Biostatistics. 2015;5: 1–6. doi:10.3844/amjbsp.2015.1.6

Reviewer #2: One additional suggestion to improve the manuscript content: line 147, add brief details of the search strategy.

Line 283: wording should be “discharge”

---

## [Author Response · Author response to Decision Letter 1]

8 Sep 2019

We are grateful for the reviewers’ comments and for the opportunity to improve the manuscript. 

Response to reviewers’

Reviewers' comments:

Reviewer #1: All my comments from the first round of the review were adequately addressed. There are only 4 minor issues left, that should be worked out prior to publication. 

Line 81: missing "to", should be ...for their discharge to home

Response: Thank you, we have added “to”.

Line 123: should be ....medicine ward specialised in gastroenterology

Response: Thank you. Corrected.

Line 321: should be "In a systematic review Beattie and colleagues identified 11 instruments measuring patient experience of healthcare quality [26]." or "Beattie and colleagues identified 11 instruments of measuring patient experience of healthcare quality in a systematic review [26]."

Response: Agree. Changed to: In a systematic review Beattie and colleagues identified 11 instruments measuring patient experience of healthcare quality [26]." (line 325-326)

And - sorry for grouching, but - the DOI for Reference 43 leads me to the wrong publication, I cannot retrieve the right publication with the given DOI 10.12788/jhm.3037 via doi.org. In the citation 43 as given I can only find another publication, were the title and one authors name match.

The DOI 10.12788/jhm.3037 given in the citation 43 leads to the following publication:

Brotman DJ, Siddiqui Z, Siddiqui Z, Durkin N. Does Patient Experience Predict 30-Day Readmission? A Patient-Level Analysis of HCAHPS Data. Journal of Hospital Medicine. 2018;13. doi:10.12788/jhm.3037

The publication you cited as No. 43 has another DOI, when I look for the title only:

Siddiqui OI. Methods for Computing Missing Item Response in Psychometric Scale Construction. Current Research in Biostatistics. 2015;5: 1–6. doi:10.3844/amjbsp.2015.1.6

Response: Thank you for your thoroughness! We agree and have corrected to:

“43. Siddiqui OI. Methods for Computing Missing Item Response in Psychometric Scale Construction. Am J Biostat, 2015. doi:10.3844/amjbsp.2015.1.6” (line 578-579)

Reviewer #2: One additional suggestion to improve the manuscript content: line 147, add brief details of the search strategy.

Response: Thank you. We agree and have added: “Our search strategy comprised the following terms: patient experience, patients satisfaction and patient perspective, and further patient discharge, patient transfer, continuity of patient care, patient hand over, patient hand off, primary health care, home based care, nursing homes, community health services and and community based care. (line 147-151).

Line 283: wording should be “discharge” 

Response: Thank you. We think this sentence was imprecise, and have added “scores on” to clarify: “The relations of scores on DICARES, and the three factors, with readmission within 30 days are shown in Table 3.” (line 287-288)

We hope that our responses have addressed the comments sufficiently and that our manuscript would be acceptable for publication.

---

## [Editor Report · Decision Letter 2]

16 Sep 2019

Discharge care quality in hospitalised elderly patients: Extended validation of the Discharge Care Experiences Survey

PONE-D-19-15237R2

Dear Dr. Boge,

We are pleased to inform you that your manuscript has been judged scientifically suitable for publication and will be formally accepted for publication once it complies with all outstanding technical requirements.

With kind regards,

Prof, Mojtaba Vaismoradi, PhD, MScN, BScN

Academic Editor

PLOS ONE,

Nord University, Bodø, Norway

---

## [Editor Report · Acceptance letter]

19 Sep 2019

PONE-D-19-15237R2 

Discharge care quality in hospitalised elderly patients: Extended validation of the Discharge Care Experiences Survey 

Dear Dr. Boge:

I am pleased to inform you that your manuscript has been deemed suitable for publication in PLOS ONE. Congratulations! Your manuscript is now with our production department. 

With kind regards,

on behalf of

Professor Mojtaba Vaismoradi 

Academic Editor

PLOS ONE